# APTrans: Transformer-Based Multilayer Semantic and Locational Feature Integration for Efficient Text Classification

Gaoyang Ji [1], Zengzhao Chen [1,2], Hai Liu [1,2], Tingting Liu [3] and Bing Wang [1,*]

1. Faculty of Artificial Intelligence in Education, Central China Normal University, Wuhan 430079, China
2. National Engineering Research Center for E-Learning, Central China Normal University, Wuhan 430079, China
3. School of Education, Hubei University, Wuhan 430072, China
* Correspondence: sxwangbing@mail.ccnu.edu.cn

**Abstract:** Text classification is not only a prerequisite for natural language processing work, such as sentiment analysis and natural language reasoning, but is also of great significance for screening massive amounts of information in daily life. However, the performance of classification algorithms is always affected due to the diversity of language expressions, inaccurate semantic information, colloquial information, and many other problems. We identify three clues in this study, namely, core relevance information, semantic location associations, and the mining characteristics of deep and shallow networks for different information, to cope with these challenges. Two key insights about the text are revealed based on these three clues: key information relationship and word group inline relationship. We propose a novel attention feature fusion network, Attention Pyramid Transformer (APTrans), which is capable of learning the core semantic and location information from sentences using the above-mentioned two key insights. Specially, a hierarchical feature fusion module, Feature Fusion Connection (FFCon), is proposed to merge the semantic features of higher layers with positional features of lower layers. Thereafter, a Transformer-based XLNet network is used as the backbone to initially extract the long dependencies from statements. Comprehensive experiments show that APTrans can achieve leading results on the THUCNews Chinese dataset, AG News, and TREC-QA English dataset, outperforming most excellent pre-trained models. Furthermore, extended experiments are carried out on a self-built Chinese dataset theme analysis of teachers' classroom corpus. We also provide visualization work, further proving that APTrans has good potential in text classification work.

**Keywords:** text classification; feature fusion; T-PTLM; semantic information; deep learning





## 1. Introduction

Text classification is one of the fundamental tasks in natural language processing (NLP), which aims to understand the meaning of text expression and determine the category of the text. This task has broad application prospects in sentiment analysis, document topic classification, spam detection, etc., and it is also the basis for other tasks of NLP. In recent years, the accuracy of this task has been greatly improved due to diverse convolution strategies [1–4], multitasking learning [5], tree loop information [6], extra-mapping relation [7], and invariant knowledge information [8]. However, challenges per-sist for real-world application when complex contexts, colloquial information, and diversified expressions are prevalent.

### 1.1. Challenges

Nowadays, deep neural networks, such as convolutional neural networks (CNNs), recurrent neural networks (RNNs), and graph neural networks, have become common for NLP tasks and are widely utilized for text classification. The Transformer-based [9]

pre-training language model [10–12] has achieved remarkable results due to the excellent ability of multi-head attention mechanisms in mining semantics. However, semantic and location information cannot be simultaneously captured. A possible reason is that a complex and deep network may struggle to simultaneously comprehend both aspects of information in a single learning session because information about location is often lost when semantic mining is conducted. In some challenging environments (Figure 1), such as colloquial information, word confusion, and chaotic word positions, the more explicit words are not adequately expressed due to information confusion, which is devastating for the existing Transformer-based approach because it requires words with accurate meanings. Accordingly, utilizing the limited accurate information available in these existing sentences is crucial for effectively mining semantic and location information, resulting in accurate predictions. Recently, several studies have delved into pre-trained downstream task models based on the Transformer architecture and achieved certain effects, mainly by using pre-trained model parameters to enrich word sense relations in the subsequent networks, thereby preserving long-dependency semantics that are nearly intact. However, in these network architectures, most of them do not consider retaining low layers' text feature information, among which location information accounts for the majority, which greatly affects the further improvement in the classification effect. Therefore, the utilization of the relationship between the position and the meaning of a word is attractive for high-precision and robust text classification research.

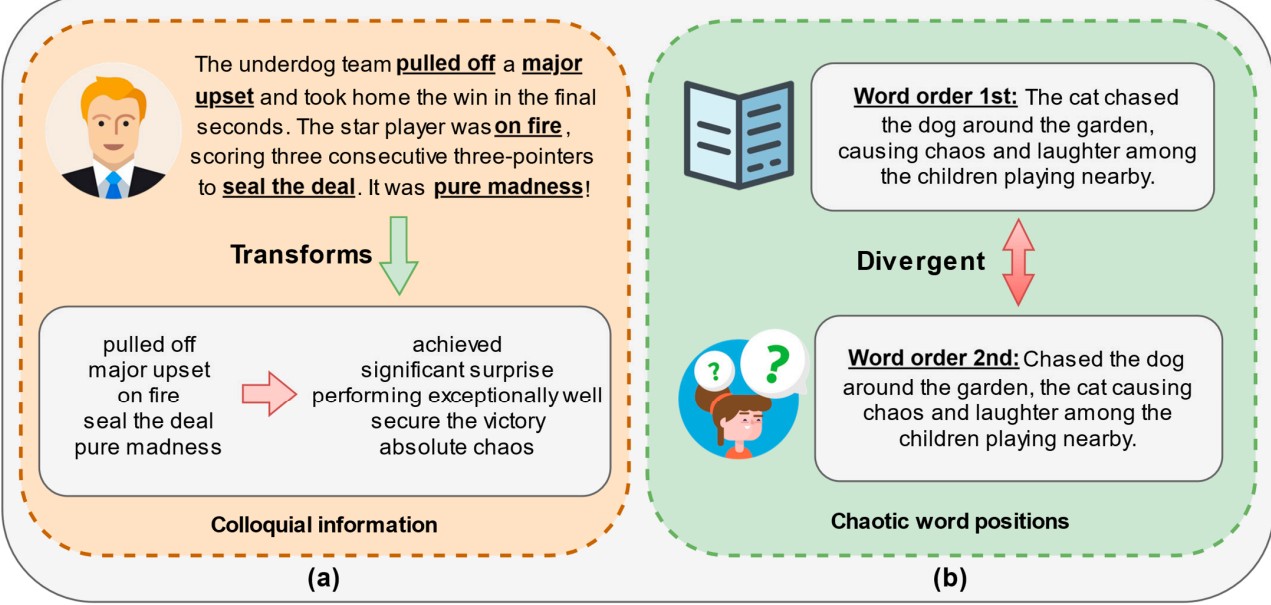

**Figure 1.** Existing challenges in text classification, including (**a**) colloquial information and (**b**) chaotic word positions. In these scenarios, the semantics are significantly difficult to understand, resulting in difficulties for text classification. In (**a**), the words in bold and underline refer to colloquial expressions, which can easily cause problems in understanding. The underlined and bold text in (**b**) indicates two sentences with different word orders, which makes it difficult to understand the semantics. The bold words at the bottom of the two figures indicate the main problem that appears in the text that this figure focuses on, which is also the theme of itself.

### 1.2. Observations and Insights

We investigate three good clues through careful observation in this study for utilizing the key location and semantic information of language to promote text classification. First, key relationships will always exist in a text. Despite the presence of colloquial information to a certain extent, we can make specific decisions by identifying core semantic information. For example, even when a sentence contains a substantial amount of colloquial information (Figure 1a), we can still capture the key meaning of the sentence, such as the words

"team" and "scoring". Grabbing special features in the image can achieve a major effect breakthrough, such as in [13–16]. Specifically, accurate prediction can be achieved despite the serious interference of colloquial information by utilizing the semantic relationships of the remaining key words. This important cue is defined as core relevance information. Second, positional associations are inherent in statements. When a sentence contains multi-semantic expressions, additional information in the sentence is invariably required to define the connotation of the term (Figure 2). This issue cannot be resolved solely through simple pre-training; rather, its necessitates a greater emphasis on grasping the correlation relationships in a sentence. Therefore, by paying attention to the locational relationship between the core words and other words in the sentence, it is helpful for the model to determine the semantic connotation of the sentence and avoid semantic confusion, thus improving the accuracy of prediction. Third, in our methodological exploration, we observed that deep-level models often excel at capturing the semantic connotation of words. However, these models seemingly overlook the position information of sentences, which could greatly affect the mining of the internal correlation relationships of statements. Although each word in a sentence can be easily understood, the different order of the words always affects the overall semantic understanding of the method (Figure 1b). The three clues we found point to positional and semantic information in statements that are necessary for efficient text classification in real-word applications.

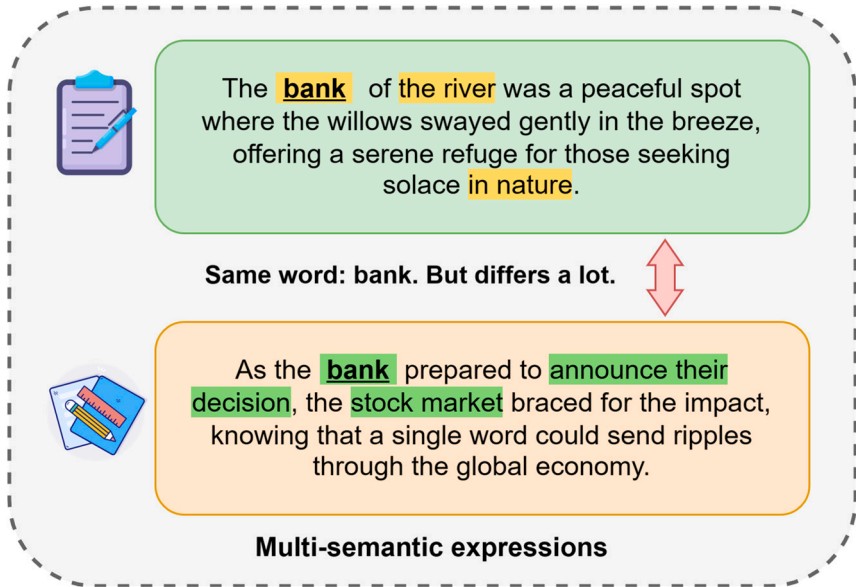

**Figure 2.** An illustration of the influence of multiple semantic expressions. The different colors in the figure represent the relationship between word groups in the text and should also be the focus of the attention mechanism. The highlighted word 'bank' in bold and italic forms different semantic associations with different highlighted words in two sentences. The different lexical relationships make the word 'bank' express different meanings.

We reveal two insights about text classification, namely, key information relationship and word group inline relationship, based on the three clues found. We believe that these two new insights about location and semantic information are enlightening for further efficient text classification applications. The two key insights are outlined below:

***Key insight I: key information relationship.*** Several key relationships exist within a particular piece of text (a statement). The few key words in a statement and the relationships between them are defined as key information relationships, which are crucial for prediction and are more powerful and reliable than patterns that solely focus on overall semantics. Figure 1a shows that the relationship between the core words in the text has rich information that affects the semantic understanding. These core words constitute the decisive factor of

prediction. The predictive harm caused by irregular expression can be greatly reduced on the basis of critical minority relationship learning.

*Key insight II: word group inline relationship.* Given that a sentence consists of many word groups, a relationship exists with a certain information between them. This property is defined as a word group inline relationship, which helps the model to determine the connotation of multi-semantic words and avoid the situation of word meaning confusion. Figure 2 shows that attention is distributed in two regions before and after, allowing for repeated mining of correlations between the two to gather more inline information for prediction. This type of relationship is often easier to be mined by the Transformer-based architecture due to its particularity. However, how to better use the characteristics of this approach to further improve efficiency remains an urgent problem to be addressed.

Given the above-mentioned key insights on statements, the question is how to design a model that can take advantage of such heuristic ideas. Traditional deep networks cannot solve this problem. By contrast, the Transformer is more effective in addressing this issue. Recently, the pre-training model of the Transformer has become a popular research topic in NLP. The Transformer has shown strong and extraordinary capabilities through the preliminary exploration of the pre-training parameters and further optimization to efficiently complete tasks. Therefore, T-PTLM should be used in the mining of key information relations, and the learned feature marks can better identify the word group inline relations. Although pre-trained models can play a certain role, it is obviously not a suitable scheme to rely solely on pre-trained models in the field of text classification, because they do not fully reflect the category characteristics of word meanings in the text classification process. Hence, it is urgent and necessary to use the two novel clues proposed by us to develop a proprietary text classification model.

*1.3. Contributions*

Based on the two key insights and T-PTLM, this study aims to achieve better text classification results by solving the problem that deep networks cannot fully extract location information from text expression and the feature fusion trouble of text location information and semantic information. Accordingly, this study proposes APTrans, a method designed to discover and utilize the internal semantic relationships and key information relationships in the text through the Transformer architecture. This method can discover the correlation between tokens by using the sub-attention method, and the core semantic features will be strengthened. Attention information learned about features can be visualized through vector similarity. Specifically, we have constructed a hierarchical feature extraction architecture. The backbone network extracts the associated information in the text and passes it to subsequent modules for processing. APTrans will fuse the semantic and positional information in the text to obtain the comprehensive representation. In summary, we have made the following contributions:

- Three clues are obtained in the survey statements, including core relevance information, semantic location associations, and the mining characteristics of deep and shallow networks for different information. The core correlation information and semantic location associations confirm that the existence of key words in the text and the relative location information have an important impact on the effect of the final classifier. The other remaining clue ensures that we consider the lower layers' information as important as the high layers' information during the development of the model. Based on this, we propose a discriminant method, APTrans, for text classification to leverage our findings and address challenging environments.

- We reveal two key insights about sentences, namely, the key information relationship and word group inline relationship. In the proposed hierarchical deeper attention module, we improve the procedural computing mechanism of multi-head attention, enhancing the ability to extract the relationship between words by parallelizing attention computing processing. Moreover, the attention calculation of the text vector output from the backbone network is carried out again to ensure that the model can

fully mine the key lexical semantic and location information in the text. The FFCon module transmits information through hierarchical iteration and shrinks text features through MLP, aiming to complete the fusion of multi-level semantic and positional information. Based on this, APTrans is able to solve the challenges caused by colloquial words or multi-semantic texts by mining and integrating the relative position information of texts, so as to achieve better classification results.

- APTrans has been shown to be efficiently applied to text classification in real NLP tasks through a large number of experiments and to perform comparisons on multiple text classification benchmark datasets. In addition, we demonstrate through modular ablation experiments that the two proposed modules, the hierarchical deeper attention module and FFCon module, have an important impact on the overall effect of the model. Furthermore, we conducted visualization experiments to prove that the model effectively leverages locational information for prediction.

The rest of this article is structured as follows: Section 2 reviews some related work of text classification. Section 3 describes the proposed method APTrans, which has a feature mapping backbone, a hierarchical deeper attention module, a feature transfer fusion FFCon module, and a compressed connection classification head module. Section 4 presents the detailed experiments and results. Section 5 concludes our work.

## 2. Related Work

### 2.1. Problem Formulation

We summarize the text classification task as follows: Given a piece of text $x$ and its corresponding category label $y$, the goal is to discover a mapping function $f$ that makes the estimated label $\hat{y} = f(x)$ as identical as possible to the actual label. Neural network technology is used to design the mapping function $f$, which is mainly about the adjustment of the network architecture. At present, numerous network architectures have emerged for text classification, such as RNNs, GCNs, and Transformer. After the network is created, the parameters in the mapping function $f$ are typically obtained by adjusting the prediction error between the label $\hat{y}$ and the true label $y$.

### 2.2. Model Based on Traditional Methods

At present, in text classification tasks, deep learning technologies have predominantly yielded good results. Traditional deep learning technologies are mainly divided into methods based on CNNs, RNNs, attention mechanisms, and other text classification models. TextCNN [1] is the earliest text classification method with CNNs. The concept of multi-channel convolution proposed by Yoon Kim uses pre-trained word vectors word2vec and multi-channel embedding technology to achieve certain results in public datasets. Johnson et al. [2] proposed DPCNN for text classification. They obtained longer semantic information through equal-length convolution and pooling and deepened the number of convolution layers to achieve significant results. To validate the effectiveness of convolutional layers for text semantic understanding, Le et al. [3] input character-level text vectors and word-level statement vectors into shallow CNNs and DenseNet. They found that the effectiveness is improved when paired with deep convolutional networks. Meanwhile, word-level statement vectors are more suitable for shallow convolutional networks. However, the biggest problem faced by CNNs is that they cannot fully extract the dependencies between longer texts. The research on RNNs has addressed the problem of long-statement dependence to a certain degree, and the development of LSTM [17] and GRU [18] has gradually increased the number of RNNs used for text classification. Liu et al. [5] designed a multi-task LSTM model for text classification with small amounts of data. Tai et al. [6] studied tree networks and designed the Tree-LSTM architecture. They verified through experiments on sentiment classification datasets that the tree structure is superior to the sequential LSTM structure to a certain extent. In [19], Zhou et al. simultaneously extracted Chinese character-level vectors and word-level vector representations, built the C-BLSTM

model, and achieved good results in short-text classification. She and Zhang [20] comprehensively used a CNN and RNN and developed a CNN-BiLSTM architecture based on skip-gram technology, which achieved certain results in text classification tasks. Khan et al. [21] also combined the advantages of a CNN and LSTM to build a network architecture capable of extracting long-term dependencies and preserving local information. Combined with machine learning classifiers, they achieved certain results in the task of classifying special language emotions. In [22], the bidirectional GRU mechanism is used for convolution layer association to adapt to corpus and feature vocabulary, thus achieving better results in emotion prediction tasks. Text classification methods based on attention mechanisms have become the focus of numerous scholars' research to further explore the dependencies between texts. A hierarchical attention architecture HAN was developed by [23], which divided documents into the sentence level and word level to calculate attention and construct different features, completing the document classification task. Zhou et al. [24] constructed a hybrid attention network HANs, simultaneously utilizing character-level and word-level attention to obtain feature vectors, and made achievements in Chinese short-text classification. In [25], Jang et al. used Bi-LSTM and a convolutional neural network to construct an additional attention mechanism, and this mixed attention mechanism performed well in mining text semantic information and achieved good generalization effects on datasets. Zheng et al. [26] built a semantic representation network with multi-attention mechanisms, which showed good effects on emotion classification tasks by mining semantic information at different levels. Similarly, the attention mechanism enables [27] us to achieve significant performance improvements in feature perception, thus achieving better discriminant ability. Apart from these methods, numerous other algorithms have attracted much attention from researchers. In [28], Yao et al. first proposed the graph convolutional network for text classification. The text classification work was converted into a graph classification work without relying on external knowledge by constructing a network graph between the word level and the document level. The model achieved good classification results. Zhou et al. [29] proposed a new model with a multi-fragment dynamic semantic technique based on a spatiotemporal graph convolution network and achieved the optimization of the method. The experimental results exceeded most of the baseline models and achieved good results. Yang et al. [30] proposed a new method HGAT based on the heterogeneous graph of documents, topics, and entities and set up a dual-attention structure to determine the significance of the different types of nodes to achieve classification. Reference [31] constructed a capsule network to maximize the utilization of the features in the invariant knowledge space for complete classification. They designed a type of iterative and adaptive cross-text classification strategy, which fully improved the computational efficiency. However, most of these methods rarely use the Transformer [9], resulting in a lack of mining of text word-level relationship dependencies in the network and poor semantic understanding.

### 2.3. T-PTLM Technology

The proposal of Transformer [9] has greatly promoted the development of NLP technology. The Transformer-based pre-trained language model (T-PTLM) combines the Transformer with self-supervised learning (SSL) [32] technology to extremely accelerate the development of NLP. Transferring the features of text representation from large-scale corpora to downstream tasks has achieved excellent results in understanding and generating tasks.

Peters et al. [33] first proposed EMLO for generating pre-trained word vectors. EMLO used a bidirectional LSTM model combined with contextual context to generate the representation of each word for downstream tasks, which triggered an exploration of the pre-training methods in NLP. With the introduction of the Transformer in the field of NLG, Radford et al. [34] developed a generative pre-training model named GPT based on the decoder architecture. This model realized the pre-train–fine-tune framework and was further optimized with the introduction of Prompt engineering technology [35], GPT-2 [36], and GPT-3 [37]. In the realm of NLU, Devlin et al. [10] introduced the BERT method

based on the encoder mechanism, which predicts contextual information using the mask mechanism to complete semantic extraction and achieve multi-directional technology to predict the current word, which is difficult to achieve in general semantic models. Based on the Bert encoder, [38] constructs a classifier to recognize emotion information in chat text and achieves good effect in terms of emotion prediction through the enhanced text standardization layer to recognize the original semantics. In [12], Yang et al. proposed a model XLNet that can extract bidirectional semantic features. They avoided the masking mechanism in the BERT model by using auto-regressive (AR) instead of auto-encoding (AE) technology. The dual-stream attention method is introduced to address the inconsistent data between pre-training and fine-tuning, and the relative position encoding and segment loop mechanism in Transformer-XL [39] are used for reference to achieve target perception and attention calculation, breaking a number of records in NLP tasks. The performance greatly exceeds that of BERT. In this study, we observe the key information relationship and word group inline relationship that exist in the text, which are difficult to explore by using traditional networks. Accordingly, we decide to use the Transformer that is more suitable for exploring long dependencies between words. The pre-trained XLNet architecture in the Transformer's library [40] is utilized as the backbone of our model to further mine fine-grained semantic association information in the text for text classification.

### 2.4. Summary of Previous Work

In general, in previous studies on text classification, the most widely used methods are traditional deep learning networks, which often cannot achieve sufficient and ideal classification effects. With the emergence and development of the Transformer [9], research on optimizing downstream classification tasks based on pre-trained models has become popular in recent years. However, these studies tend to focus on capturing the text semantic vector of the last layer of the backbone, and ignore the feature information of low layers' feature computation, which is considered to have a significant impact on the model classification effect in our previous analysis. These features contain more relative position information and some semantic information, which is helpful to solve the semantic confusion in the process of classification. In this study, we mainly propose a classification method for the contraction and fusion of text hierarchical information, APTrans, which treats the features at the lower layers equally with the features at the higher layers, ensuring that the proposed method can integrate the text location information and effectively distinguish semantic confusion problems so as to achieve better classification performance.

## 3. Proposed Method

### 3.1. Architecture Overview

APTrans is a hierarchical attention text semantic analysis method based on feature fusion. The details of our model are shown in Figure 3. The architecture mainly contains four parts: feature mapping backbone, hierarchical deeper attention module, FFCon module, and compressed connection classification head. Specifically, the feature mapping backbone is used to extract fine-grained and multi-dimensional information in sentences. In this study, we mainly adopt XLNet [12] as the backbone network because it efficiently performs in exploring the relationships between long texts in semantic understanding problems using a masking mechanism and a permutation and combination semantic analysis scheme. Moreover, we made modifications by introducing a masking mechanism in the input mapping layer to enhance the effectiveness of semantic extraction. The second part is the hierarchical deeper attention module, which is used for extracting the deep and multi-scale correlation information of each layer of text mapping vectors. The further processing and extraction of this module can effectively deepen the understanding of semantic information in the low layers' network, and avoid the semantic confusion leading to the poor classification effect. The third part is the FFCon module, which transfers and integrates the fine-grained features of the text from the higher layers to the lower layers, and effectively completes the fusion of the semantic features and the location information.

This ensures that the proposed method attaches equal importance to the two parts of the information and facilitates the improvement in the classification performance. The fourth part is the compression connection classification header. The feature information after pooling is connected and used as text extraction features for the final classification.

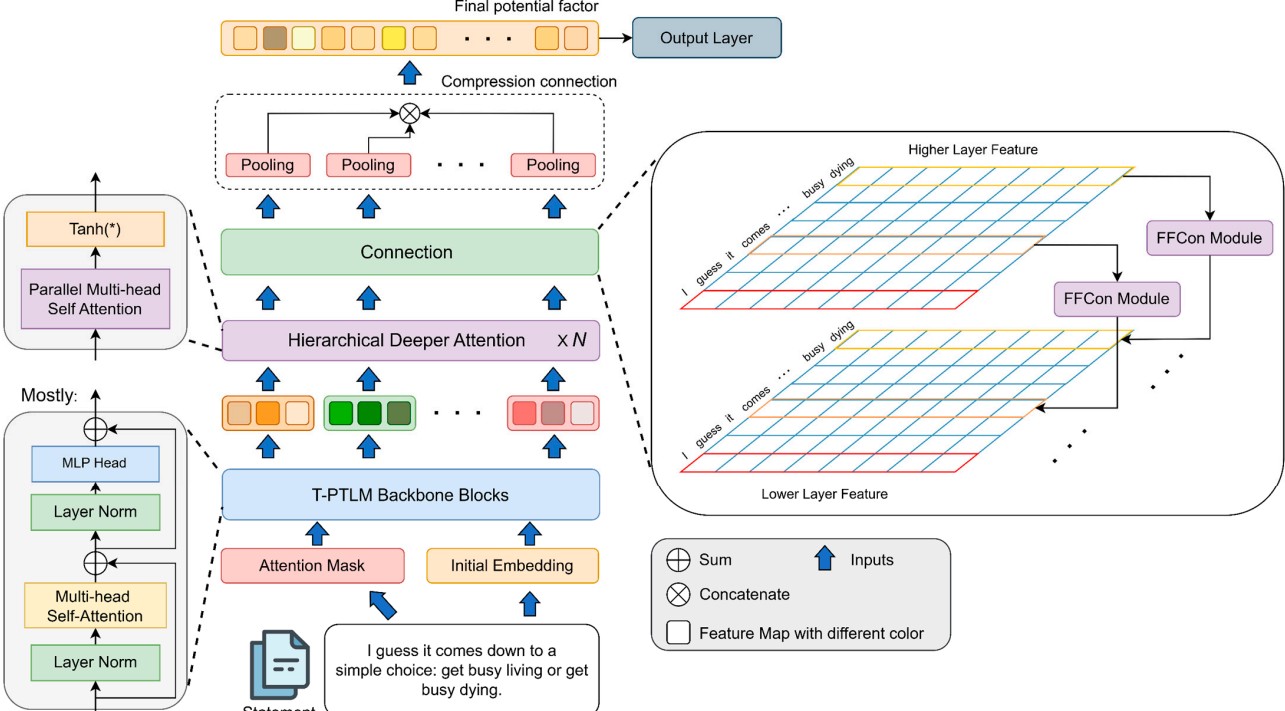

**Figure 3.** The architecture of APTrans. First, the statement will be masked and go through the embedding block before entering the T-PTLM backbone. Second, T-PTLM will hierarchically output text vectors to the hierarchical deeper attention module for further mining of long-distance dependencies. The matrices of different colors in this process in the figure represent the feature relationships mined from the text at different levels. Thereafter, the vector is layered into the connection block, which is mainly composed of the FFCon module to complete the fusion of high-level semantics and low-level location information. The compression connection module is then inputted to complete the pooling of semantic vectors. Finally, the classification header predicts the resulting text category.

### 3.2. Feature Mapping Module

The feature mapping module processes the statement and generates the feature vector. This process can be divided into three stages: preliminary loading of statements into index vectors, word vector mapping, and traversing the Transformer block. After processing, the original text sentences are transformed into specific vectors for computation.

*Phase I: initial load index vector.* At this stage, the input text is segmented according to the length of the sentence and projected into a 1D vector. First, the input text sentence $x \in R^{Seq}$ will be segmented by the word segmenter according to unit words and will be segmented according to the sentence length or supplementary sentences to generate new sentences. The segmentation process is as follows:

$$x' = Cut(x), \ x' \in R^{Seq'}, \tag{1}$$

where $x'$ represents the new sentence after segmentation or supplementation, and $Seq'$ denotes the length of the new sentence. This value is artificially determined. Thereafter, a vocabulary index 1D vector is generated based on each word. An attention mask index 1D vector is also generated, which will map each word $w_i, i \in 1, 2, 3, \cdots, Seq'$ into a specific vector. The index vector of the text is expressed as $v_0 \in R^{Seq'}$, which is mainly composed of the index of the vocabulary. If one word does not appear in the vocabulary, then its

index is replaced by the index of [*UnK*]. Each attention mask index vector of a sentence is represented as $m_0 \in R^{Seq'}$. This vector mainly consists of two numbers, 0 and 1, where 1 indicates that the word is a supplementary word and should not be considered for attention calculation, while 0 means that the word is an original word and attention calculation should be performed.

*Phase II: vector mapping.* Given that the Transformer layer requires a certain semantic vector input, at this stage, the index vector obtained through processing is used to load the 2D feature vector of each word. Specifically, the text statement is mapped into an exact semantic vector through a dictionary. The projection formula is as follows:

$$c_i = E(v_i), \ i \in 1,2,3,\cdots,n, \tag{2}$$

$$p_0 = [c_1, c_2, c_3, \cdots, c_n], \tag{3}$$

where $c_i \in R^E$ represents the vector of each word after projection; $p_0 \in R^{Seq' \times E}$ represents the matrix containing the word vector, that is, the semantic matrix of the statement; $Seq'$ represents the sentence length; and $E$ represents the dimension of the vector after mapping each word. This process can be accomplished by pre-training a dictionary of word vectors.

*Phase III: traverse transformer block.* The semantic feature vector will go through M Transformer blocks. Each Transformer block will go through dual-stream attention calculations by using the attention mask index vector of the sentence and finally generate an implicit semantic vector. The formula is as follows:

$$q_0^l = BinaryAttention(p_0, m_0), l \in 0,1,2,\cdots,M, \tag{4}$$

where $BinaryAttention()$ represents the attention calculation mechanism in the backbone network architecture, $p_0$ represents the semantic feature vector of the text statement, $m_0$ represents the mask index vector generated in the first stage, and $q_0^l \in R^{Seq' \times E}$ represents the implicit semantic vector generated after the text with sequence number 0 passes through the $l$ th Transformer layer. In summary, there will be a total of M Transformer modules, generating M layers of implicit semantic vectors. Finally, the obtained multilayer implicit semantic vectors are hierarchically inputted to the next module to continue the feature extraction work.

### 3.3. Hierarchical Deeper Attention Module

The semantic association information extracted at this time is often insufficient to represent the multi-scale meaning of the text at this stage due to the different number of attention processing layers of these implicit semantic vectors mined by backbone attention computing. The hierarchical deeper attention module is used to utilize lower-level semantic representations and further mine the associated information in the semantics.

Assuming the presence of M stages in the backbone network architecture, and that each stage will output an implicit semantic representation vector $q_i^l, l \in 0,1,2,\cdots,M$, $i \in 0,1,2,\cdots,N$, represents the implicit semantic representation vector obtained after the $i$th paragraph of text passes through the $l$th stage. This semantic vector needs to be further processed. The process is divided into $n$ same stages to mine semantic information. The detailed calculation steps of each stage are as follows:

First, the semantic vector processed by each layer should go through the attention calculation again. The process is expressed as follows:

$$\begin{cases} Q_{act}^0 = q_i^l, \\ Q^t = Attention(Q_{act}^0), t \in 0,1,2,\cdots,n, \\ Attention(q_i^l) = f(Q = q_i^l, K = q_i^l, V = q_i^l), \\ f(Q,K,V) = Softmax\left(\frac{QK^T}{\sqrt{d_k}}\right)V, \end{cases} \tag{5}$$

where $Q^t \in R^{Seq' \times E}$ represents the semantic vector obtained by the attention calculation processing of the $t$th stage, and $Q^t_{act}$ represents the final output after activating the $t$ th stage. $Q^0_{act}$ is initialized as the implicit semantic vector hierarchically outputted by the backbone network. The *Attention*() function is the same as the attention mechanism in the Transformer. The multi-head attention calculation implemented in this network architecture is a parallel computing architecture and a non-serial computing mode, ensuring the improvement in model calculation efficiency. The normalized exponential function *softmax*() is an activation function:

$$Softmax(z_i) = \frac{e^{z_i}}{\sum_{c=1}^{C} e^{z_c}}, \tag{6}$$

where $z_i$ is the result of the $i$th node's calculation, while $C$ denotes the number of output nodes, namely, the quantity of categories for classification.

After the attention calculation is completed, activation processing is performed. This process can be expressed as follows:

$$Q^t_{act} = \tanh(Q^t), \tag{7}$$

where tanh represents the hyperbolic tangent function:

$$\tanh(z) = \frac{e^z - e^{-z}}{e^z + e^{-z}}. \tag{8}$$

The model will retain the initial mapped text vector $p_0 \in R^{Seq' \times E}$ as an input into the deep attention calculation. In the last stage, the model will no longer perform activation processing and only perform attention calculations. After passing through the hierarchical deeper attention module, the text representation vector $Q^n \in R^{Seq' \times E}$ will be obtained. We can obtain multi-scale and multi-dimensional fine-grained semantic information by further mining the hierarchical implicit semantic information in the backbone network, which facilitates subsequent models to understand text semantics.

*3.4. FFCon Module*

Using only the latent semantic vectors of the last layer of the backbone often ignores part of the coarse-grained word-level semantic information, causing the model's effect on text understanding to be affected. However, relying solely on hierarchical attention features may lead to certain problems. The lower-layer network only retains better-position information, but the text's semantic information has a coarser granularity. Therefore, conveying semantic information from higher to lower layers or completing the fusion of locational information and semantic information will become a crucial aspect.

The feature pyramid network [41] is a typical feature fusion network, which better fuses the high- and low-resolution information of the image to achieve good recognition results. We propose the FFCon module to effectively reduce the problem of coarse-grained semantic analysis in the lower layers of the backbone network during the semantic analysis process and to effectively integrate location information.

Assuming that we have obtained the text representation vector $Q^n \in R^{Seq' \times E}$ calculated by the hierarchical deeper attention module, the process of inputting the FFCon module is shown in Figure 4.

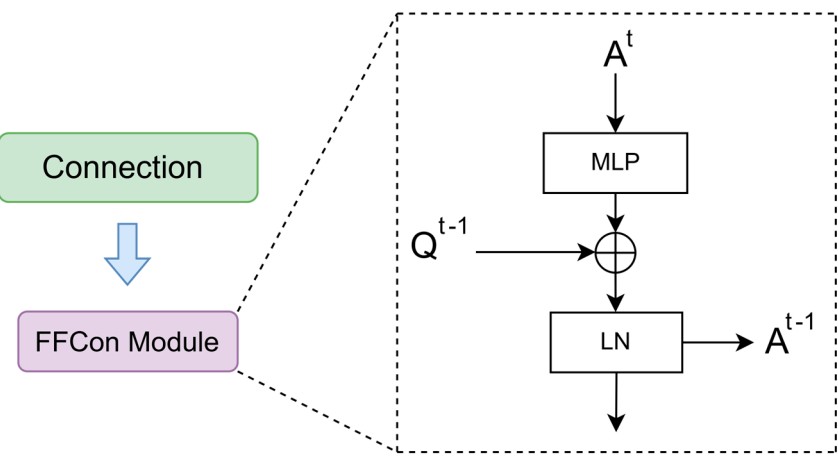

**Figure 4.** The calculation process of the FFCon module. FFCon is the key component of the connection.

First, after the connection block receives two layers of text representation input, the calculation process can be expressed as follows:

$$A^n = Q^n, \tag{9}$$

$$A^{t-1} = \mathrm{LN}\left(Q^{t-1} + \mathrm{MLP}\left(A^t\right)\right), \ t \in 0,1,2,\cdots,n, \tag{10}$$

where $Q^{t-1}$ represents the low-level semantic vector, $A^t$ represents the high-level semantic vector after passing through the connection block, and $A^{t-1} \in R^{Seq' \times E}$ denotes the result of the $t-1$th layer after passing through the connection block, which is also the semantic vector passed down to the lower layer. $A^n = Q^n$ means that the highest-level semantic vector will not be modified and will continue to be retained and transmit information to the lower levels. *LN* indicates layer normalization. MLP refers to multiple fully connected feed-forward networks, consisting of two linear transformations, including the Mish activation function [42] in the middle, which can be expressed as follows:

$$\mathrm{MLP}(x) = \mathrm{Mish}(x \cdot W_1 + b_1)W_2 + b_2, \tag{11}$$

$$\mathrm{Mish}\ (z) = x \times tanh(\ \ln(1 + e^x)), \tag{12}$$

where $W_1$ and $W_2$ are linear projections, and $b_1$ and $b_2$ represent the deviations in the fully connected neural network. Different parameters are used between layers. The inner layer dimension is larger than the input and output dimensions. The semantic information is mapped to a high-dimensional space and then to a low-dimensional space, extracting deeper layer features. This processing method, similar to the computing mechanism in the Transformer [9], makes the core information of the statement in the high layers' fusion features more explicit after shrinking and Mish activation, and enlarges the important influence factor, which is conducive to subsequent feature fusion and classification. In this network, the input and output dimensions are as follows: $d_{model} = 768$, and the dimension of the inner layer is $d_{ff} = 768 \times 3$. Therefore, our model can learn fine-grained and comprehensive representations in the text by further shrinking and integrating high-level semantic information into lower-level features.

*3.5. Compression Connection Module*

Compression connection integrates semantic information as features extracted from the text for final classification. After receiving the semantic vector $A^t \in R^{Seq' \times E}$, $t \in 0,1,2,\cdots,n$ from the FFCon module, mean pooling is first utilized on these semantic vectors for dimensionality reduction. We can denote the process as follows:

$$A^t_{pooling} = \varphi\left(A^t\right), \tag{13}$$

where $\varphi$ represents the mean pooling operation, and $A^t_{pooling} \in R^E$ represents the overall semantic characteristics of the text after pooling. Thereafter, features from all stages are concatenated into an aggregated semantic vector. The calculation formula is as follows:

$$A = \left[ A^1_{pooling}, A^2_{pooling}, \cdots, A^n_{pooling} \right],$$ (14)

where *A* represents the aggregated feature. Thereafter, the feature is fed to the classification head to produce the prediction result $\hat{y}$. The entire process can be summarized as follows:

$$\hat{y} = A \cdot W + b,$$ (15)

where *A* represents the aggregated feature, *W* is the linear projection, and *b* represents the bias in the linear neural network.

In this work, a certain amount of text sentences $X = \{x_1, x_2, x_3, \cdots, x_r\}$ and their true category labels $Y = \{y_1, y_2, y_3, \cdots, y_r\}$ are determined for training. The objective is to discover the best parameters $\theta$ through the maximum likelihood estimation to enable the model to have outstanding classification performance. Meanwhile, the CrossEntropy function is utilized to calculate similarity between the predicted distribution and the actual label. The overall loss computing process is as follows:

$$Loss = -\sum_{t=1}^{r} y_i \cdot \log\left( \hat{y}_t \right) + \varepsilon \|\theta\|^2,$$ (16)

where $\varepsilon$ refers to the parameter of *L*2 regularization, which has a great contribution to mitigate overfitting. At last, the BERTAdam optimizer is employed to minimize the target loss function.

## 4. Experimental Results

The platform environment for our experiments is the Windows 11 64-bit system, equipped with a Nvidia RTX A40 GPU (Nvidia, Santa Clara, CA, USA), 64 GB running memory, Cuda 11.3 parallel computing platform, and a PyTorch 1.13.1 framework for deep learning. Model training is implemented in Python 3.11.

### 4.1. General Setting

#### 4.1.1. Datasets

Three classic datasets (THUCNews, AG News, and TREC-QA) are introduced to test the effects of APTrans. Both THUCNews and AG News datasets belong to news datasets. THUCNews is a Chinese news dataset and AG News is an English news dataset. The distribution of data in the two datasets is uniform and there is no imbalance problem. The reason for choosing these two datasets is that there is a long textual relationship in the news text, and there are some non-written expressions such as colloquial expressions. Experiments on these two datasets can verify the learning ability of APTrans for long-text information and determine whether our method can capture key lexical location information in the text, so as to achieve a good classification effect. The TREC-QA dataset belongs to a short-text dataset in Q&A research. The problem of uneven distribution of category data exists in this dataset, but the typical feature of samples in this dataset is that there are always core words in each sample to summarize the semantic information of the sample. Therefore, we used this dataset for training and testing to judge whether APTrans has the capacity for short-text classification and whether it can achieve high-performance classification by grasping core words in short texts under the condition of uneven data. More details about these three datasets are shown below:

**THUCNews** [43]: A tremendous Chinese text classification dataset contains approximately 840,000 news files and is divided into 14 categories. This dataset was created by screening the historical data of Sina News RSS subscriptions from 2005 to 2011. We

used stratified random sampling to extract the dataset utilized in our work. The training dataset comprises 40,000 texts, the validation dataset consists of 8000 texts, and the test dataset exhibits 2000 texts, covering a total of 10 categories, including sports, entertainment, property, and education. Chinese sentences of varying lengths can be found in the corpus, and the semantic complexity of the Chinese corpus is high, bringing a certain degree of difficulty to the processing of text classification.

**AG News** [44]: AG News (AG's News Corpus) is a news dataset that contains four categories for topic classification. Each category contains a total of 30,000 training samples and 1900 test samples. This dataset was originally derived from a public corpus database of AG's articles. Moreover, this dataset contains titles and content of news, resulting in several irregular colloquial expressions, making it difficult for the model to achieve accurate semantic understanding and classification.

**TREC-QA** [45]: This dataset is a question dataset that contains approximately 6000 common English questions and is widely used in Q&A research. This dataset is mainly divided into two versions: **TREC-6** and **TREC-50**, both of which have a training dataset with 5452 questions and a test dataset with 500 questions. We used **TREC-6** in this study, which contains six categories of coarse labels, including people, locations, digital information, and other issues. Most of these texts are short texts that may pose certain challenges to the semantic analysis capabilities of the model.

### 4.1.2. Evaluation Metrics

The commonly used text classification evaluation metric is the accuracy of the test dataset. The classification accuracy measure can be written as follows:

$$\text{Accuracy} = \frac{1}{q} \sum_{i=1}^{q} \text{J}\left(y_i, \hat{y}_i\right),\tag{17}$$

where $\text{J}()$ refers to the discriminant function, $\hat{y}_i$ denotes the label of the $i$th statement predicted by the model, and $y_i$ represents the actual label of the $i$th statement. The formula of the discriminant function is as follows:

$$\text{J}(a,\,b) = \begin{cases} 0, \text{ if } a = b, \\ 1, \text{ if } a \neq b. \end{cases}\tag{18}$$

### 4.1.3. Compared Methods

We introduced several classic Transformer-based methods for comparative experiments to verify the effectiveness of our model APTrans. The following mainly introduces some representative pre-training model methods. In specific experiments, we also referenced models with better performance on corresponding datasets, so as to enrich experimental data, ensure the fairness of comparison experiments, and prove the scientificity and effectiveness of our method.

**BART** [46]: Lewis et al. proposed a special text generation model. Specifically, this model combines the characteristics of BERT and GPT-2 [36], adds random noise to the text to destroy the text, and allows the model to self-learn how to restore the original text, building a powerful generative architecture that efficiently performs in NLU.

**ALBERT** [47]: A simple but effective network architecture, this model greatly reduces the number of parameters required by proposing two mechanisms, factorized embedding parameterization and crosslayer parameter sharing. Using a self-built self-supervised loss function that focuses on the consistency between sentences allows the model to have better downstream task performance.

**RoBERTa** [11]: This model achieves better downstream task performance by improving the pre-training research of BERT, removing the next prediction loss, applying a dynamic mask mechanism, and increasing the training time and sequences.

**ERNIE** [48]: Sun et al. integrated AR and AE networks, used phrase-based and entity-based masking mechanisms, adopted incremental training and continual learning mechanisms, and expanded the training scale. Therefore, this model achieves better semantic understanding and efficiently performs text classification on Chinese datasets.

**ELECTRA** [49]: Clark et al. proposed a replacement token detection study. Specifically, the input is destroyed by replacing the text with the generator, and the discriminative model is trained to determine whether a token has been replaced in the text, thereby achieving better results than the masking language modeling (MLM) pre-training method.

**MPNet** [50]: They proposed a novel method for training, which combines the advantages of BERT and XLNet, utilized the dependency between the predicted tags through permutation language modeling, and introduced auxiliary location information as an input, which reduces position difference during the model training process. The effect is better than MLM and PLM.

*4.2. Implementation Details*

In this study, we completed the experiment according to the following details and steps. In the model architecture, first of all, due to the limitations of the experimental conditions, the backbone network we adopted contains 12 layers of attention parameters from shallow to deep. Therefore, combined with the initial embedding semantic vector, a total of 13 layers of semantic vector information will participate in the subsequent calculation. Secondly, in the hierarchical deepening attention module, the parallel attention computing module has three layers. It is worth mentioning that when the last layer completes the attention computing, the module does not activate the semantic value again, so as to retain the semantic information mined. Finally, the FFCon module transmits a total of 12 layers of semantics and locational information from higher to lower layers, completes feature fusion through effective semantic reduction MLPs, and then participates in classification through pooling connections. The initial weights (in this study, the addresses of the pre-training parameters loaded by the backbone network in all experiments are as follows: English pre-training parameter address: https://huggingface.co/xlnet/xlnet-base-cased, accessed on 30 May 2024; Chinese pre-training parameter address: https://huggingface.co/hfl/chinese-xlnet-base, accessed on 30 May 2024) of the backbone are loaded with the officially trained English parameters of XLNet [12] and the Chinese parameters trained by [51] based on a Wikipedia dump [10]. The remaining neural network parameters are randomly initialized for training.

In terms of training parameter setting, first, the statements entered into the model are cropped to a specified and uniform size to fairly compare the effects. The clipped sentence size hyperparameter is set comprehensively after the maximum sentence length statistics and average sentence length calculation of the dataset to ensure that different sentence length parameters are set for the long-text dataset and the short-text dataset to make the experimental model understand the text semantics scientifically. We trained this model for 100 epochs by using the BERTAdam optimizer with a weight decay of 0.001. For the training process on long-text datasets, the batch size was set as 16, the learning rate $\alpha$ was 0.00005, and the sentence length was arranged as 512. In terms of short-text training, the batch size was 64, $\alpha$ was $5 \times 10^{-5}$, and the sentence length was 128. In particular, when the effect does not improve, the learning rate will drop tenfold, to as low as $5 \times 10^{-6}$ on short-text datasets and as low as $5 \times 10^{-7}$ on long-text datasets. This is known as the learning rate multiplier decay strategy. Furthermore, we used an early stopping strategy for all models to avoid model overfitting. The early stop strategy means that when the learning rate reaches the minimum and the model effect on the test set is not improved after a long time of training (in our experiments, we set the time to 500 steps, and 1 step is equal to 10 batches), the program will automatically stop the training process and retain the best model training parameters. In the experiment, in order to ensure the comparability of the baseline model and our proposed method, we used the same hyperparameter settings and training methods for all models on the same dataset. In terms of the selection of specific

hyperparameters in the experiment, we set the fixed statement length hyperparameters after several attempts based on the ability of the GPU we have. Regarding the choice of learning rate and weight decay, we determined different hyperparameters during short- and long-text training through many experiments. During the experiments, we tried different learning rates, such as $5 \times 10^{-3}$, $5 \times 10^{-4}$, and $5 \times 10^{-5}$, and we also tried different weight decay rates, such as 0.001, 0.01, and 0.1. However, we found that these two hyperparameters had little impact on the final model effect due to the existence of the early stop strategy and the learning rate multiplier decay strategy; so, we chose the value with a higher efficiency for the learning rate and the value suggested in the official document for weight decay. Therefore, the set hyperparameters are reasonable. All experiments were performed using the PyTorch toolbox and an Nvidia RTX A40 GPU.

### 4.3. Experiment Results and Analysis

In our work, in order to ensure the fairness of the comparison experiment, all of the comparison methods we used were guaranteed to have the same model architecture and we set the same architectural parameters as those in the original literature. In addition, we fine-tuned our selected dataset locally based on a baseline model loaded with complete pre-trained parameters from https://huggingface.co/, accessed on 30 May 2024, to ensure the best results on the dataset. At the same time, the method compared by the model has advanced significance in the current development process of text classification. For example, on the Chinese dataset, the effect of ERNIE is basically equivalent to that of the best model; on the English datasets, XLNet and BART can approximate the best generalization results in classification tasks. Finally, for all of the methods, we adopted the scheme of five equal weights averaged and took the average accuracy obtained through five experiments as the final classification effect of the model, thus avoiding the influence of randomization parameters. Therefore, this ensures that the experimental data are real and effective and shows that the optimization effect of the proposed module is clear. In the following, we compare our approach to these excellent models and analyze the performance of different approaches. The top performing values are shown in **bold**, and those following the top values by the underlining.

### 4.3.1. Results on the THUCNews Dataset

We used several advanced pre-training model methods in NLP to conduct experiments on the THUCNews dataset. The detailed comparison results are shown in Table 1. All of the pre-training models on display are based on the Transformer, and all of them load the relevant Chinese pre-training parameters. The training process maintains the same strategy (e.g., learning rate decay) to ensure the fairness of comparison. All methods are based on the Transformer architecture, but the models show diverse effects on the task due to the different architectures of the models themselves. Although T-PTLM [32] has iterated numerous methods, BERT [10] still achieves good results on the training set, which may be due to the reinforcement of the pre-training parameters. ERNIE [48], as a Chinese pre-training method, also achieves good results on the task because it combines different training strategies, enabling the model to continuously expand and adjust parameters. XLNet [12] achieves impressive performance on the task, indicating that two-stream attention is effective enough. In comparison with these methods, APTrans has achieved the best results on this task, with the best accuracy on the validation and test datasets (Dev: 98.55%, Test: 97.89%). This result indicates that our method successfully uses the key information relationship and the inline relationship of word groups in the text, realizes the fusion of location information and semantics, and achieves good results.

**Table 1.** A performance comparison among the APTrans approach and Transformer-based methods on the THUCNews dataset.

| Methods | Pad Size | Dev Acc (%) | Test Acc (%) |
|---------|----------|-------------|--------------|
| BigBird [52] | | 96.85 | 95.73 |
| ALBERT [47] | | 96.95 | 96.64 |
| DeBERTa [53] | | 97.15 | 96.33 |
| ELECTRA [49] | | 97.45 | 97.32 |
| GPT-2 [36] | | 97.80 | 97.51 |
| BART [46] | 512 | 97.85 | 97.10 |
| RoBERTa [11] | | 98.00 | 97.11 |
| BERT [10] | | 98.30 | 97.67 |
| ERNIE [48] | | 98.30 | 97.15 |
| XLNet [12] | | 98.45 | 97.67 |
| APTrans (ours) | | **98.55** | **97.89** |

The top performing values are shown in **bold**, and those following the top values by the underlining.

### 4.3.2. Results on the AG News Dataset

We introduced MPNet [50] and other methods on the English dataset to continue the experiments. APTrans was also compared with these advanced methods on the AG News dataset following the same principle of training rules to maintain fairness. Table 2 shows the results of the comparison. On this long-text dataset, our method still achieves advanced results, reaching an accuracy of 94.68%. SHGCN [54] connects the relationships among the text, entities, and words to complete the information transfer among each other by building a heterogeneous graph convolutional network, which also achieved results on this dataset. However, since no pre-training model was introduced, the model could still be further improved and developed. MPNet achieves good results on this news dataset by integrating training strategies and adding auxiliary location information. Our approach goes a step further in harnessing the diversity of information in the text, with location information proven to improve accuracy. The ability of the T-PTLM approach to achieve expressive performance compared with traditional models is primarily due to the Transformer's ability to leverage long-term dependent semantic relationships hidden between all patches. Overall, APTrans performs most efficiently among the methods listed in Table 2. Our approach improved the accuracy by 0.11% compared with the pre-trained baseline architecture. This result shows that our model effectively extracts and fuses a small number of core association relationships and semantic locational association representations in the text.

**Table 2.** A comparison on the AG News dataset between APTrans and other typical methods.

| Methods | Pad Size | Acc (%) |
|---------|----------|---------|
| SHGCN [54] | None | 88.38 |
| ERNIE [48] | | 90.79 |
| DeBERTa [53] | | 91.38 |
| ALBERT [47] | | 93.13 |
| MPNet [50] | | 93.22 |
| GPT-2 [36] | 512 | 94.20 |
| ELECTRA [49] | | 94.28 |
| RoBERTa [11] | | 94.42 |
| XLNet [12] | | 94.51 |
| BERT [10] | | 94.57 |
| APTrans (ours) | | **94.68** |

The top performing values are shown in **bold**, and those following the top values by the underlining.

### 4.3.3. Results on the TREC-6 Dataset

We conducted a comparative experiment on the TREC-6 dataset to verify whether our proposed method is equally valid on short-text datasets. Under consistent training rules, the obtained results of the experiments are shown in Table 3. The quantitative consequence shows that APTrans still achieves the best results in short-text classification. The implementation data show that in several methods, such as BERT [10], short text has a great influence on the prediction effect of the model. After the introduction of the replacement token detection training strategy, ELECTRA [49] achieves a higher prediction level. BART [46] creates a new generative architecture by adding random noise, which also has excellent performance. This notion indicates that the random noise fluctuation of the text is also conducive to the improvement in the results. In addition, our APTrans can fully mine the text location information when the text is short and understand the dependency of these statements. Our model achieved a 0.6% improvement compared with the traditional baseline methods. It shows that our model can fully extract semantic information from a corpus and integrate location information, understand semantics, and be competent for short-text classification.

**Table 3.** Comparative results with some classical methods on TREC-6 dataset.

| Methods | Pad Size | Acc (%) |
|---|---|---|
| ERNIE [48] | | 86.20 |
| ALBERT [47] | | 95.80 |
| GPT-2 [36] | | 96.40 |
| DeBERTa [53] | | 96.80 |
| BERT [10] | 128 | 97.00 |
| RoBERTa [11] | | 97.20 |
| ELECTRA [49] | | 97.40 |
| BART [46] | | <u>97.40</u> |
| APTrans (ours) | | **98.00** |

The top performing values are shown in **bold**, and those following the top values by the <u>underlining</u>.

### 4.3.4. Ablation Study

We performed ablation studies on our APTrans model. Considering the factor of time efficiency, we only conducted experiments on the AG News and TREC-6 datasets, and the parameter settings, such as sentence length, are the same as those in the previous experiments. Ablation experiments were performed on hierarchical deeper attention and FFCon to verify the validity of these two modules. The effects of the approaches were ascertained by experimenting with different collocations of these two modules. Figure 5 shows our experimental results. The two modules significantly improve the performance of APTrans. On the AG News dataset, the hierarchical deeper attention module improves the accuracy by 1.54% compared with the APTrans without any modules. Under the same comparison conditions, the FFCon module shows a great improvement in accuracy (94.22%), and the cooperation of the two modules results in the increase in the value by 1.80%, reaching 94.68%. Moreover, the addition of the two modules also greatly improves the prediction accuracy on the TREC-6 dataset, reaching 98.00%. The data from the modular ablation experiment fully demonstrate the obvious advancement of our two proposed modules, which shows that learning the key information relationship and word group inline relationship can indeed help improve the accuracy of the method. This also fully shows that the ideas and innovations we put forward are meaningful.

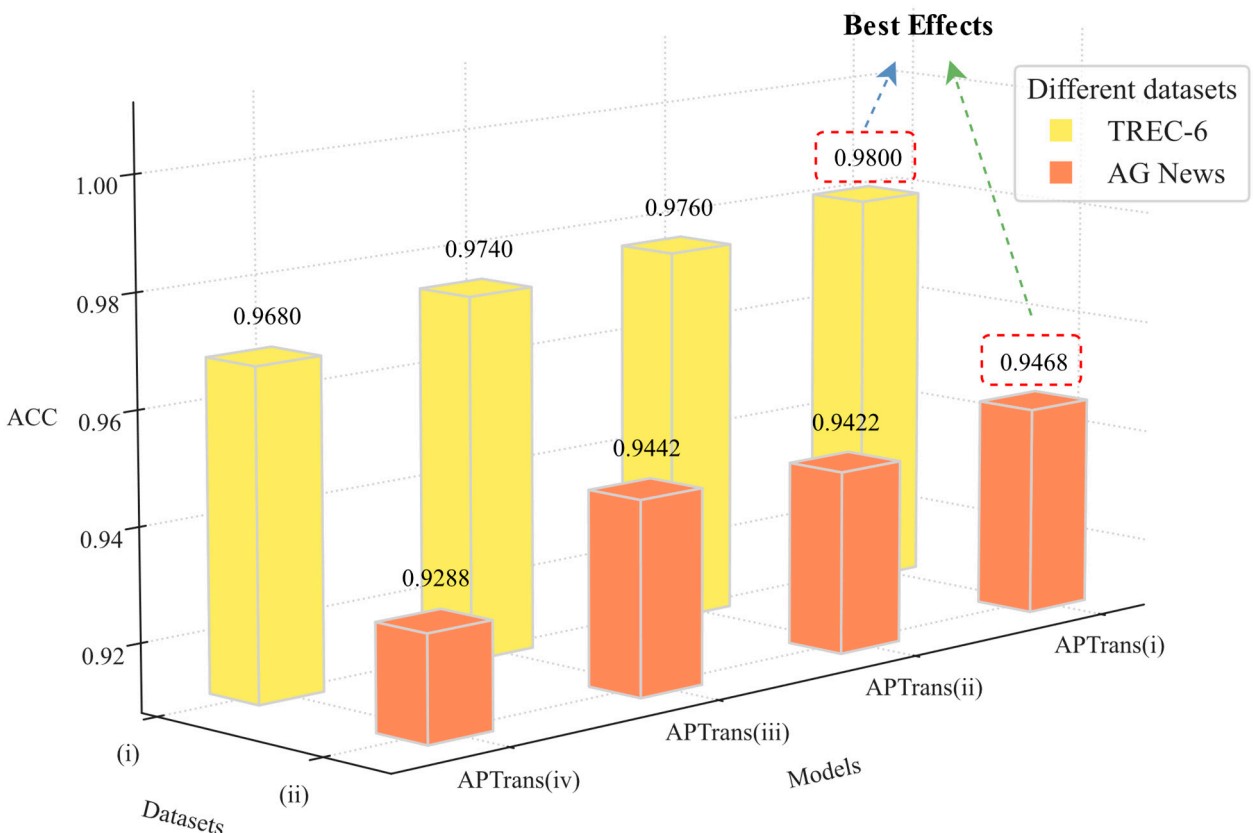

**Figure 5.** Ablation studies on the hierarchical deeper attention module and the FFCon module. We conducted experiments on two datasets: (i) the *TREC-6 dataset* and (ii) the *AG News dataset*. The experimental results on four different types of collocation combinations demonstrate the importance of the two modules. APTrans(i): *our model with the two modules*; APTrans(ii): *APTrans without hierarchical deeper attention*; APTrans(iii): *APTrans without FFCon*; and APTrans(iv): *APTrans without the two modules*.

### 4.3.5. Confusion Matrix and Analysis

We can summarize and visualize the performance of the classification algorithm through the confusion matrix. The number on the diagonal represents the correct prediction, and the others are the wrong prediction. The higher the percentage value shown diagonally, the more accurate the algorithm. This metric can reflect the tolerance degree of the algorithm to easily distinguish confused data.

In this experiment, two datasets, TREC-6 and THUCNews, were selected for visual analysis, and five or six methods were used for comparison on each dataset (Figures 6 and 7). Figure 6 shows the accuracy of different methods on various categories of data, which are extracted from the confusion matrix calculation process. In Figure 7, if the diagonal color is darker, then the prediction accuracy of the corresponding category is higher. The actual label is on the *X*-axis, and the prediction label is on the *Y*-axis. Among all of the methods, our model (Figures 6 and 7f) shows strong classification ability because it mines the key information relations in sentences and the word group inline relations to learn the special distinguishing information in the text.

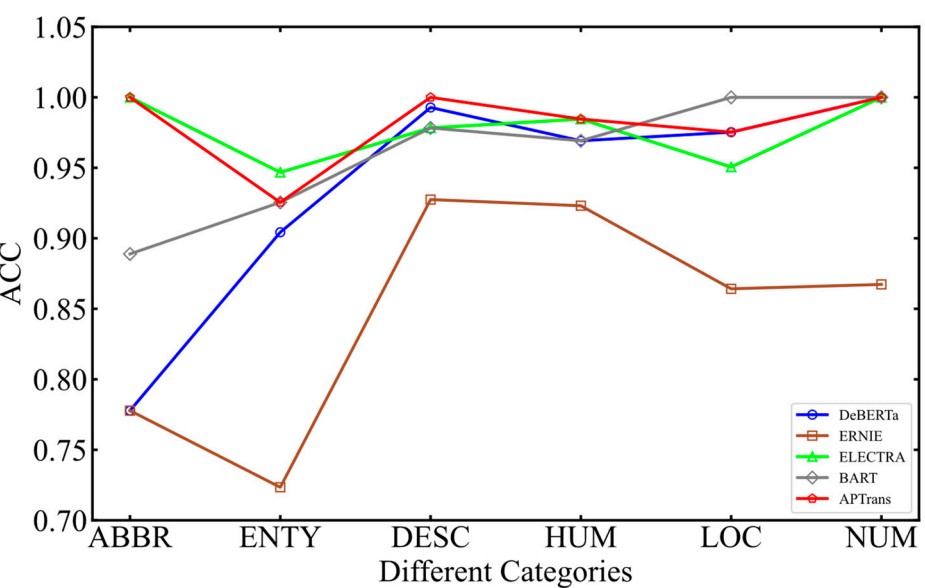

**Figure 6.** Visualization on TERC-6 dataset. Line chart for four different types of methods. ABBR: *abbreviation*. ENTY: *entity*. DESC: *description and abstract concept*. HUM: *human being*. LOC: *location*. NUM: *numeric value*.

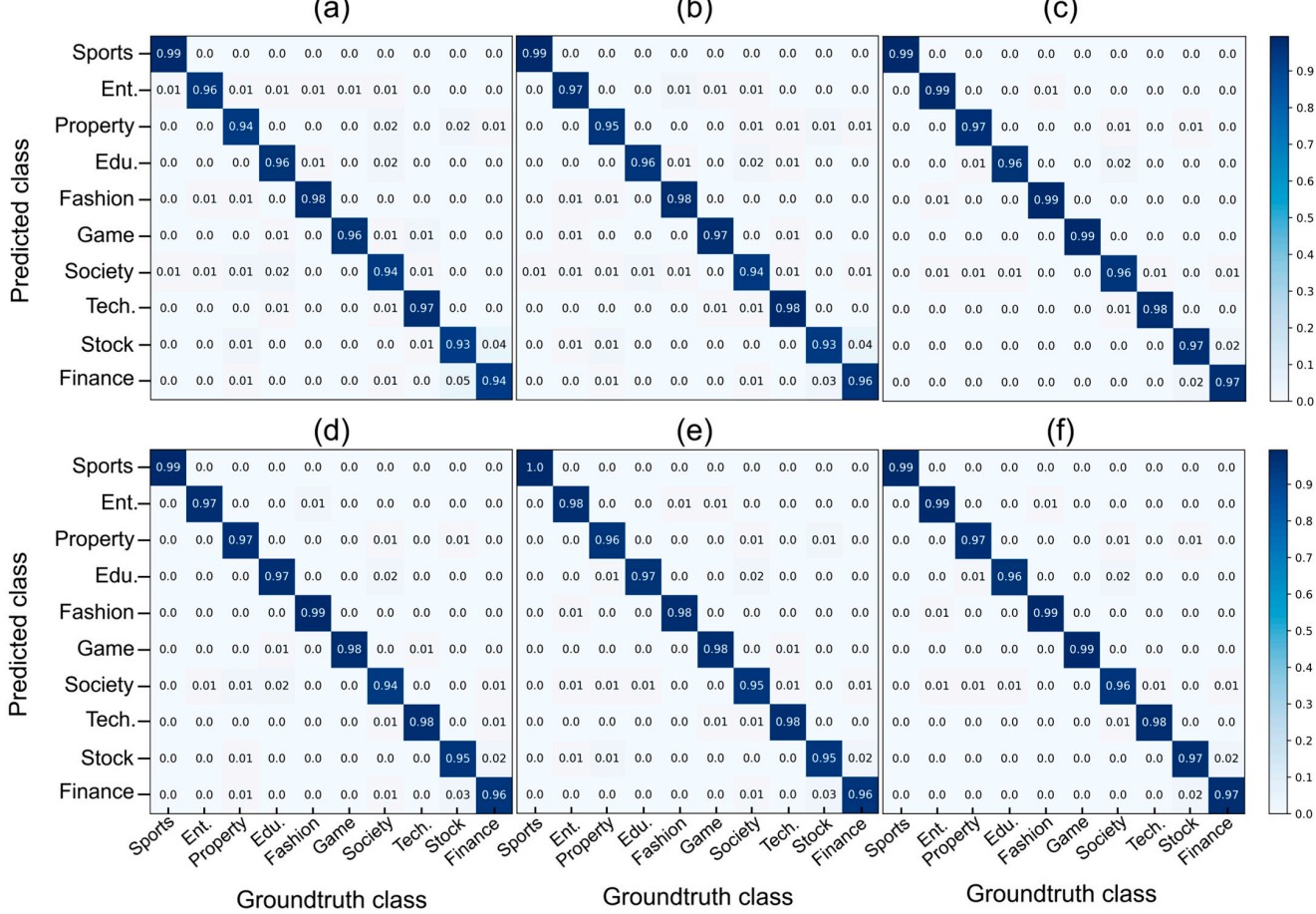

**Figure 7.** Visualization on THUCNews dataset. Confusion matrix for six different types of news. (**a**) BigBird. (**b**) DeBERTa. (**c**) GPT-2. (**d**) RoBERTa. (**e**) ERNIE. (**f**) APTrans (ours). Ent.: *entertainment*. Edu.: *education*. Tech.: *technology*.

### 4.3.6. Expansion Experiments

In text classification applications in real life, taking into account colloquial expressions, the corpus of teachers' teaching in the classroom meets our research needs. In the classroom, the colloquial expressions of teachers and students will intersect in the classroom corpus because their interaction will have an important influence on the teaching effect. On this basis, we established the teacher classroom corpus theme analysis dataset TATC. This dataset comes from actual engineering classroom corpus data and has typical classroom teaching characteristics, such as teachers asking questions and waiting for answers. The dataset covers four categories, such as scientific and technological literacy, and contains about 4000 manually annotated corpora certified by experts. This dataset is divided into training and test datasets, with a division ratio of 0.8. Figure 8 shows the relevant information of TATC.

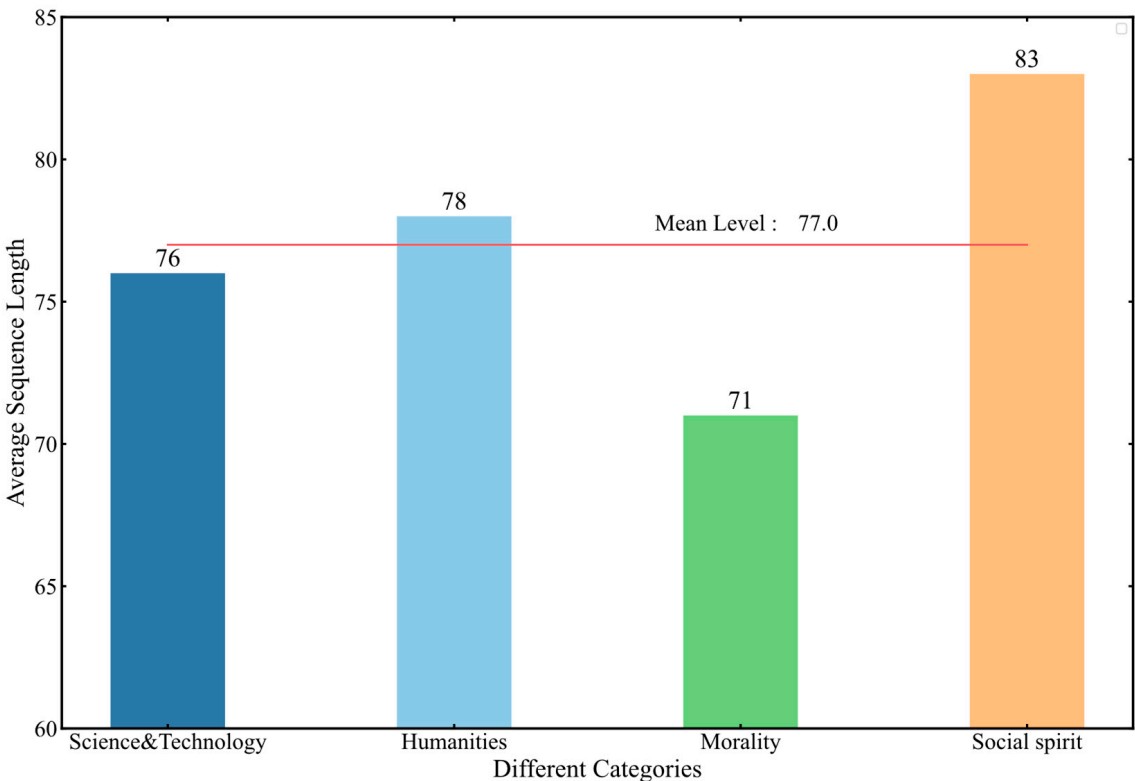

**Figure 8.** Basic information of the TATC dataset. The dataset contains four categories, and the evaluation sentence length for each category and the total average sentence length of the text are shown in the figure above.

We utilized APTrans to conduct extended experiments on the TATC dataset. Table 4 shows the quantitative results. The results show that our method still has strong competitiveness compared with the preliminary training model based on the Transformer. The accuracy on the self-built dataset can reach 83.33%, exceeding the XLNet baseline model by 1.30%. Although our method is built for written text, the comparison results show that APTrans can indeed fully capture the semantic information in the text and fuse the position information to achieve strong results. This notion further means that our method can be applied in practice, proving the universality of our model.

**Table 4.** A comparison with some T-PTLM methods. The results of our experiments are presented as follows.

| Methods | Pad Size | Acc (%) |
|---|---|---|
| BigBird [52] | | 77.34 |
| DeBERTa [53] | | 73.69 |
| ALBERT [47] | | 73.70 |
| ELECTRA [49] | | 80.47 |
| GPT-2 [36] | 128 | 82.02 |
| XLNet [12] | | 82.03 |
| RoBERTa [11] | | <u>83.06</u> |
| BART [46] | | <u>83.07</u> |
| ERNIE [48] | | <u>83.07</u> |
| APTrans (ours) | | **83.33** |

The top performing values are shown in **bold**, and those following the top values by the <u>underlining</u>.

## 5. Conclusions

In this study, we propose a text classification method APTrans based on a hierarchical attention pyramid architecture to cope with the problem of being unable to fully extract text position information and word association information at the same time. We reveal the key information relationship and word group inline relationship in statements. To take advantage of these features in the text, the XLNet model is first selected as our backbone to extract long-dependency semantic information and coarse-grained location information in sentences. The mapping semantics of embedding and the implicit semantic variables of each layer in the backbone are extracted as inputs, and attention computing is further used to mine associated semantic information. Thereafter, the feature fusion module FFCon is established to input semantic information of higher layers to the lower layer for fusion calculation with coarse-grained position information to obtain multilayer implicit semantic features from the text. Finally, the compression connection module of APTrans compresses and cuts the hierarchical features and completes the feature stitching input classification header to calculate the label corresponding to the text. We test APTrans on multiple text classification benchmark datasets. The experimental consequence clearly shows that our model, APTrans, can efficiently integrate word position information and semantic information in the text and achieve good results in text classification. Meanwhile, the success of APTrans illustrates the importance of lexeme relationships in texts, which have probably been overlooked in previous studies. We hope that our preliminary study can stimulate further research on the positional and semantic relationship of words in NLP and further promote the development of this field.

**Author Contributions:** Conceptualization, G.J. and Z.C.; methodology, G.J. and Z.C.; validation, G.J.; formal analysis, G.J., Z.C. and H.L.; investigation, G.J., Z.C. and H.L.; resources, G.J. and Z.C.; data curation, G.J., Z.C. and H.L.; writing—original draft preparation, G.J.; writing—review and editing, G.J., Z.C. and H.L.; visualization, G.J.; supervision, Z.C. and H.L.; project administration, G.J., T.L. and B.W.; funding acquisition, Z.C., H.L, T.L. and B.W. All authors have read and agreed to the published version of the manuscript.

**Funding:** This study was supported by the National Natural Science Foundation of China (grant nos. 62277026, 62377037, 62277041, 62211530433, 62177018), in part by the National Natural Science Foundation of Hubei Province project (nos. 2022CFB529, 2022CFB971), the Jiangxi Provincial Natural Science Foundation under grant (no. 20232BAB212026), the university teaching reform research project of Jiangxi Province (grant no. JXJG-23-27-6), and the Shenzhen Science and Technology Program under grant no. JCYJ20230807152900001.

**Institutional Review Board Statement:** Not applicable.

**Informed Consent Statement:** Not applicable.

**Data Availability Statement:** The datasets used in this article are available upon request from the corresponding author.

**Conflicts of Interest:** The authors declare no conflicts of interest.

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
