# Peer review of "APTrans: Transformer-Based Multilayer Semantic and Locational Feature Integration for Efficient Text Classification"

_applsci, doi:10.3390/app14114863_

Round 1

Reviewer 1 Report

Comments and Suggestions for Authors

The authors propose an interesting direction to modify existing transformer-based models to address attention hierarchically. This idea sounds interesting. 

However, the empirical evidence supporting their claim is rather weak. In particular, the accuracy of the proposed model is only marginally higher than that of previous models. This small difference is more likely due to the choice of hyperparameters.

In their experiments, they fixed the weight decay (0.001), batch size (16), and learning rate (5e-5) for one configuration (4.2 Implementation Details). This is a very limited configuration to draw any meaningful conclusions. I would recommend to vary each hyperparameter for at least three values and run a gird search to measure the accuracies. Then compare the accuracies of the models with the best hyperparameters for each model.

Comments on the Quality of English Language

Need to fix many typos.

Reviewer 2 Report

Comments and Suggestions for Authors

In this research paper, the authors propose the APTrans architecture for text classification. However, several aspects could be enhanced:

- Providing more details about the experimental settings, such as the data split mechanism and the hyperparameters used for each model, would enable better comparison.

- The comparison tables indicate a minimal difference between the XLNet network and the proposed model. Given that the proposed method is based on the XLNet architecture, this raises questions about the efficacy of the changes proposed atop the XLNet architecture.

- All presented results appear to be obtained solely by the authors. It would be beneficial to compare the approach with results from other recent approaches tested on the same datasets to provide a broader context for evaluation.

Reviewer 3 Report

Comments and Suggestions for Authors

The article is on text classification using APTrans. This study focuses on integrating semantic and locational features for efficient text classification, addressing challenges such as diverse language expressions and inaccurate semantic information. By identifying core relevance information and mining characteristics of deep and shallow networks, this research offers valuable insights into key information relationships and word group inline relationships.

Although the article is interesting and brings valuable asset to this subject, I think some minor aspects should be improved:

1. A lot of abbreviations appear in the Abstract that should be explained already in the Abstract.

2. I highly recommend Authors to go through this IEEE guide for math equations to improve the mathematic equations presented in the paper: https://conferences.ieeeauthorcenter.ieee.org/wp-content/uploads/sites/8/IEEE-Math-Typesetting-Guide-for-LaTeX-Users.pdf

For example, equation (5) uses the variable Q in a bold font, but in the description below it, it is Q in a plain font. Is this a deliberate procedure?

Other example: tanh (not Tanh) is a function and should be in plain font not in cursive.

What is the difference between J and J?

3. Figure 4 uses different notation for sum than Figure 3.

4. I think this paper should be cited in this paper:

Thomas Wolf, Lysandre Debut, Victor Sanh, Julien Chaumond, Clement Delangue, Anthony Moi, Pierric Cistac, Tim Rault, Remi Louf, Morgan Funtowicz, Joe Davison, Sam Shleifer, Patrick von Platen, Clara Ma, Yacine Jernite, Julien Plu, Canwen Xu, Teven Le Scao, Sylvain Gugger, et al.. 2020. Transformers: State-of-the-Art Natural Language Processing. In Proceedings of the 2020 Conference on Empirical Methods in Natural Language Processing: System Demonstrations, pages 38–45, Online. Association for Computational Linguistics.

5. Description of Table 1 is on other page than the actual table.

6. Authors should emphasize their impact in this paper. For example in line 481, they write about APTrans, but is not exactly clear that it is their original solution.

Round 2

Reviewer 2 Report

Comments and Suggestions for Authors

Thank you for incorporating the necessary adjustments into the article. I recommend accepting it in its current form for publication.

Comments on the Quality of English Language

Minor editing of English language required